# Challenges and Achievements in the In Vitro Culture of *Balantioides coli*: Insights into the Excystation Process

**DOI:** 10.3390/pathogens14080725

**Published:** 2025-07-23

**Authors:** Alexandra Ibañez-Escribano, Lorena Esteban-Sánchez, Cristina Fonseca-Berzal, Francisco Ponce-Gordo, Juan José García-Rodríguez

**Affiliations:** Department of Microbiology and Parasitology, Faculty of Pharmacy, Complutense University, 28040 Madrid, Spain; alexandraibanez@ucm.es (A.I.-E.); lorees01@ucm.es (L.E.-S.); crfonseca@ucm.es (C.F.-B.); jjgarc01@ucm.es (J.J.G.-R.)

**Keywords:** *Balantioides coli*, in vitro culture, excystation process

## Abstract

*Balantioides coli* is the only ciliate currently described as an intestinal parasite of humans, although it can also infect other animals, particularly pigs. Its in vitro cultivation remains challenging, and no axenic culture system is currently available. Cultures are initiated by adding small amounts of feces containing cysts or trophozoites to the culture medium. Implantation success is lower when starting from cysts, and the mechanisms and early events of excystation remain poorly understood. In this study, we describe the sequence of events involved in excystation and identify factors potentially important for culture establishment. Cysts were obtained from orangutan feces and genetically confirmed as *B. coli*. Only viable cysts, determined by trypan blue or methylene blue exclusion, were used. After artificial digestion with pepsin and trypsin, cysts were incubated at 28 °C for up to 72 h in DMEM supplemented with L-glutamine, yeast extract, fetal bovine serum, and starch granules. Excystation began with a fissure in the cyst wall, allowing for bacterial entry. This appeared to stimulate the trophozoites, the increased motility of which progressively weakened and ruptured the wall, allowing for their emergence. Wall rupture and bacterial entry were critical for activation., whereas starch type had no apparent influence. Excystation occurred within the first hours; otherwise, cysts degenerated.

## 1. Introduction

In vitro culture of parasitic protists is a cornerstone for advancing our understanding of their biology, enabling detailed investigations of life cycle stages, pathogenic mechanisms, metabolism, and drug responses. Unlike studies relying solely on clinical samples or in vivo observations, controlled culture systems permit the manipulation of environmental conditions, the long-term maintenance of strains, and the continuous production of biological material in enough quantities and purity (i.e., free from contaminating bacteria and/or host tissues) for morphological, molecular, biochemical, immunological, and pharmacological analyses [1,2]. In the case of *Entamoeba histolytica* and *Giardia duodenalis*, the development of axenic cultures is essential for elucidating cell biology, oxidative stress responses, and gene regulation during encystation and excystation [3,4,5], as well as for sequencing the complete genomes of these parasites [6,7,8]. Thus, robust and reproducible in vitro cultivation methods are not only indispensable for fundamental research in parasitology but also provide the foundation for clinical, therapeutic, and epidemiological applications in the study of pathogenic protozoa.

*Balantioides coli* (=*Balantidium coli*) is the only ciliate currently described as an intestinal parasite of humans, although it can also infect other non-human primates and domestic animals, particularly pigs, which are considered the main reservoir of this parasite [9]. Among human parasitic protists, *B. coli* holds a unique historical and biological significance: it was the first intestinal protist described in humans, originally named *Paramecium coli* by P.H. Malmsten in 1857; it is the largest in size (trophozoites can reach up to 100 µm in length, and cysts may exceed 60 µm in diameter) [10,11]; and although infections are rare, it can occasionally have medical relevance, causing outbreaks and potentially leading to life-threatening dysentery [9,12]. Despite these features, *B. coli* remains largely understudied at the biochemical, immunological, and genetic levels [13]. It is currently regarded as an emerging pathogen and is included among the foodborne parasites prioritized for the development of control measures [14,15].

The lack of data on the physiology, biochemistry, and genetics of *B. coli* is closely linked to the challenges associated with its in vitro study. Axenic culture protocols for other intestinal and cavitary protozoa were developed more than five decades ago (e.g., *Entamoeba invadens* [16], *Entamoeba histolytica* [17], *Giardia duodenalis* sensu lato [18], and *Trichomonas vaginalis* [19]), and since then, the respective methodologies have been refined and optimized. However, in the case of endoparasitic ciliates, numerous challenges currently persist in their in vitro maintenance [20], primarily due to their apparent dependence on bacteria for survival [21]. Barrett and Yarbrough [22] were the first to establish xenic cultures of this ciliate. Since then, various monophasic and biphasic culture media have been tested, but always under xenic conditions, with an accompanying and undefined microbiota [23]. The methods developed over the past decade represent modifications of earlier procedures, utilizing different saline solutions as base media and employing various sera as sources of proteins and amino acids [24,25,26,27]. However, the results remain heterogeneous among isolates, in terms of both the success rates for culture establishment and the duration of culture maintenance [24,25,26].

*Balantioides coli* cultures are initiated by adding a small amount of fecal material containing parasite forms (trophozoites and/or cysts) to the medium [23,24,25,26,27], followed by microscopic observation to detect trophozoites; if present, subcultures are performed. To date, no studies have been conducted on the development of cysts or on the events occurring during the initial hours of cultivation. Accordingly, the aim of the present study is to describe the sequence of events involved in excystation and identify factors potentially important for culture establishment.

## 2. Materials and Methods

### 2.1. Collection and Initial Processing of Samples

Ten fecal samples were collected from orangutans (*Pongo pygmaeus*) housed at the Zoo Aquarium of Madrid. The samples were obtained non-invasively by zookeepers, who collected fresh feces from the ground of the animals’ enclosures in sterile containers early in the morning as part of routine health monitoring procedures conducted by the zoo’s veterinary staff. No animals were captured, handled, or subjected to any intervention for the purposes of this research. All procedures complied with the institution’s animal welfare guidelines, ensuring minimal disturbance to the animals throughout the process.

Samples were transported to the laboratory within two hours of collection and processed immediately upon arrival; they were never refrigerated at any stage. In the laboratory, a small portion (approximately 5 g of feces) was first examined to confirm the presence of *B. coli* cysts and to assess the possible presence of other parasites. For this purpose, the sodium acetate–ether concentration method [28] was used. Briefly, feces were homogenized in 30 mL of acetoacetic buffer (1.5% sodium acetate, 0.36% glacial acetic acid in distilled water) and filtered through a metal sieve. The filtrate was then mixed with diethyl ether in a 1:1 ratio and vigorously shaken. The mixture was centrifuged at 1500 rpm for 2 min. After removing the organic phase and part of the aqueous phase, the sediment was resuspended and examined under a microscope (Olympus BX51; Olympus, Tokyo, Japan) at 10–20× objective magnification using temporary wet mounts.

All samples tested positive for *B. coli* cysts and were selected for further processing. A fecal suspension was prepared by homogenizing approximately 2 g of fresh feces in 10 mL of distilled water in 50 mL tubes. This suspension was kept at room temperature and used as the starting material for subsequent analyses (Section 2.2 and Section 2.5) conducted in this study.

### 2.2. Cyst Quantification and Viability Assessment

Initial quantification and assessment of cyst viability were performed by mixing 100 µL of the fecal suspension with 400 µL of aqueous solutions of either 1% trypan blue or 0.1% methylene blue; in both cases, cysts stained dark blue were considered non-viable, whereas those that remained unstained or stained light blue were considered viable (Figure 1). Six samples showing over 75% cyst viability (determined by observing a minimum of 30 cysts per sample) were selected for further analysis. Cyst quantification was carried out using a Neubauer counting chamber; only samples with a concentration exceeding 5 viable cysts per µL were included in subsequent experiments.

### 2.3. Genetic Identification of the Isolates

Genetic analysis of the cysts was performed using one of the selected samples employed in the assays in order to confirm parasite species identity. Twenty microliters of the sediment obtained from that sample via the acetate–ether concentration method were diluted with 100 µL of distilled water in a watch glass, and 20–30 cysts were isolated from the concentrate using a modified Pasteur pipette under a stereomicroscope (Olympus SZH) at 10–20× objective magnification. The cysts were transferred to 0.5 mL Eppendorf tubes and washed twice with distilled water.

DNA extraction was carried out on cysts manually isolated from the sample; individual cysts were pipetted and then processed as a pooled sample following the protocol described in [29] for individual cysts. PCR amplification of the ITS1–5.8S rRNA–ITS2 region was performed as already described [30]. PCR products were visualized on a 1% agarose gel using Pronasafe (Condalab, Torrejón de Ardoz, Spain) as the intercalating agent and a UV transilluminator (NuGenius; Syngene, Cambridge, UK). After purification with the QIAquick^®^ PCR Purification Kit (Qiagen, Hilden, Germany), the amplicons were sequenced using an ABI Prism 3730XL DNA Analyzer (Applied Biosystems, now ThermoFisher Scientific, Waltham, MA, USA). Resulting chromatograms were examined using ChromasPro version 2.1.10 (Technelysium Pty Ltd., South Brisbane, Australia), and the sequences were compared against those available in GenBank using BLAST (https://blast.ncbi.nlm.nih.gov/Blast.cgi (assessed on 5 April 2025)).

### 2.4. Culture Media

Dulbecco’s modified Eagle’s medium (containing 4.5 g/L glucose, without L-glutamine and sodium pyruvate) (DMEM; Corning–Mediatech, Manassas, VA, USA) was supplemented with 20 mM L-glutamine (0.1%), 5% yeast extract, 20% heat-inactivated fetal bovine serum (30 min at 56 °C), and 50 µg of potato starch powder (Sigma, St. Louis, MO, USA). The pH of the final medium (“complete DMEM”) was adjusted to 7.0. After preparation, complete DMEM was stored at 4 °C for no longer than 24 h and warmed to 28 °C prior to use.

### 2.5. Excystation Procedure

To simulate the digestive process in the gastrointestinal tract of a natural host, cysts were subjected to enzymatic digestion with pepsin and trypsin prior to inoculation into the culture medium. One hundred microliters of the sample suspension were added to a Pyrex tube containing 2 mL of pepsin solution (0.5% pepsin, 2000 FIP-U/g; Merck KG&A, Darmstadt, Germany; 0.7% HCl and 0.9% NaCl in complete DMEM, pH 1.75). Tubes were incubated at 37 °C and 200 rpm for 30 min in a G24 environmental incubator shaker (New Brunswick Scientific Co., Edison, NJ, USA; now part of Eppendorf, Hamburg, Germany). After incubation, the tubes were centrifuged at 1500 rpm for 2 min using a Mixtasel BL centrifuge (JP Selecta, Barcelona, Spain), and the supernatant was discarded. Then, 2 mL of trypsin solution (1% trypsin, 2000 E/g; Merck, in complete DMEM, pH 8.0) were added, and the tubes were incubated again at 200 rpm for 15 min under the same conditions. Following this second centrifugation and the removal of the supernatant, 6 mL of complete DMEM were added to each tube. No antibiotics were added to the medium in order to avoid disrupting the accompanying microbiota. Cultures were maintained at 28 °C (following [26]) for a maximum of 72 h.

The appearance and development of the parasite were evaluated by microscopic examination of the sediments at each stage of the initial processing, at the onset of the cultures, and at 3, 12, 24, 48, and 72 h post-inoculation (hpi). Cysts and trophozoites were photographed and video-recorded using an Olympus DP23 camera mounted on an Olympus BX51 microscope with a 100× objective lens.

## 3. Results

All ten fecal samples analyzed were macroscopically normal, with formed consistency, normal coloration, and no visible mucus, excess liquid, or apparent parasitic forms. In all cases, ciliate cysts, but not trophozoites, were present. Viability, as assessed by vital staining, ranged from less than 10% to nearly 100%. Genetic analyses confirmed the species as *B. coli*, sequence variant A0, showing 100% identity with sequence JQ073373 (from a *Pan troglodytes* isolate), as well as with sequences OM423709 (from a *Pongo abelii* isolate) and MT252055 (from a pig isolate). Mixed sequences were detected, as evidenced by secondary base signals in the ITS1 region at positions (from the start of the ITS1 region) 32Y (major peak T, minor C), 35R (T/G), and 80Y (C/T); in the ITS2 region at position 275W (T/A); and by a single-base insertion (an additional T at position 361) near the end of the ITS2 fragment, leading to a one-nucleotide displacement and, consequently, mixed peaks in the final positions of the chromatogram (Appendix A). The alternative sequence showed 99.15% identity with several *B. coli* sequences from pig isolates, for example OR397831.

Excystation was observed in the cultures of four out of the six samples used, with free trophozoites appearing at 12 hpi. The progression is shown in Figure 2 and Appendix A. After the initial enzymatic digestion, most cysts retained a normal appearance, with the cyst wall apparently unaltered, as evidenced by the exclusion of vital staining (Figure 2A). At 3 hpi, the cyst wall appeared altered in some cysts, allowing entry of the vital dye without staining the internal cell (Figure 2B). By 12 hpi, activated cysts and free trophozoites were observed. In the activated cysts, bacteria had entered the cysts (Figure 2C,D), and the enclosed trophozoites exhibited ciliary movements and began to move slowly in a circular pattern. This movement gradually increased, and the trophozoites swam rapidly within the cysts, resulting in cyst wall distension (with an increased space between the wall and the trophozoite; Figure 2E) and elastic deformation of the cyst wall (Figure 2F,G). The repeated deformation contributed to the enlargement of a wall rupture (Figure 2G), which was eventually used by the trophozoite to emerge (Figure 2H). Once outside the cyst, trophozoites swam actively in all directions and eventually began feeding on the starch granules present in the medium (Figure 2I).

## 4. Discussion

This study presents, for the first time, a detailed analysis of the excystation process in *B. coli*, providing data that may be useful for the development of protocols aimed at promoting excystation and the initiation of cultures. Cultures were maintained for a maximum of 72 h without subculturing; therefore, the long-term effects of the excystation protocol on culture survival could not be assessed. In other studies, when trophozoites were observed 24 h after initiating culture from cysts, approximately half of the isolates were maintained in culture for up to 24 months [24].

The composition of the culture media used in other studies for the initiation and in vitro maintenance of *B. coli* has been highly variable. The first report of successful cultivation was by Hilkenmann [31], who indicated that this ciliate could grow in distilled water with 0.5% blood. However, these results should be interpreted with caution, as although the images provided in the publication seem consistent with *B. coli,* the reported occurrence in unusual samples such as urine, and especially blood, throws considerable doubt on the organism he was studying [22]. The first reliable xenic cultures were obtained by Barret and Yarbrough [22] and later confirmed [32] using the same medium previously employed for *E. histolytica* [33]. Since then, a variety of media have been used, combining different base solutions (saline, Ringer, Locke, HSre, TYSGM-9, Pavlova, DMEM) and protein/amino acid sources (human, bovine, equine, or guinea pig serum, egg, coconut water) in either dissolved (monophasic) or coagulated (biphasic) forms [22,24,25,26,27,34,35,36]. With varying success, these media have supported the establishment of cultures inoculated exclusively with cyst-containing feces, suggesting that the composition of the medium may not be a critical factor for excystation. However, a limitation of the present study is that we only tested DMEM as the base medium (previously validated for *B. coli* [26]) for inducing the excystation process.

Barret and Yarbrough [22] considered partial anaerobiosis important for *B. coli* culture, although other authors have reported that this ciliate tolerates high oxygen levels [37,38]. This parameter is not typically considered when establishing cultures, although bacterial growth and the oxidation of fecal components may lead to anaerobic or microaerobic conditions. It is clear, however, that such oxidation, along with nutrient depletion by bacteria, results in medium acidification. The negative impact of low pH on trophozoite viability has been recognized since the earliest cultivation attempts [32] and may explain why trophozoites are not generally considered infective [39,40]. Nonetheless, experimental infections with trophozoites [34] and epidemiological data in humans [41] and great apes [42] suggest that infection from trophozoites may still be possible. While no specific studies have been carried out with cysts, they must be capable of surviving the acidic gastric environment to initiate intestinal infection. Our implementation of artificial digestion using a pepsin-HCl solution followed by trypsin incubation resulted in a high percentage of trophozoite-positive cultures (>60%), exceeding the success rates reported in studies that inoculated media directly with cyst-containing feces [24,25,26]. This increased success may be due to enzymatic action contributing to initial cyst wall weakening. However, since many cysts retained their typical morphology during the first hours post-inoculation and excluded vital dyes (indicating an intact cyst wall), wall rupture does not appear to result solely from enzymatic action. It is more likely that a combination of factors weakens the cyst wall, eventually allowing for bacterial entry through fissures.

A combination of enzymatic digestion and mechanical forces may act on the cyst wall in vivo within the intestine of the host to produce such fissures. Once activated, the rapid movements of the trophozoite inside the cyst likely generate repeated minor deformations, gradually enlarging these fissures. In vivo, intestinal content mixing may also promote cyst rupture through physical collisions with other debris. In our system, we attempted to simulate this process by agitating the complete DMEM containing starch particles. However, this system may have been suboptimal, as the less dense cysts may have remained in suspension during agitation while the denser starch granules settled at the bottom of the tubes. Further experiments are needed to optimize this setup and assess the role of mechanical rupture in cyst activation.

Beyond its potential role in cyst rupture, starch is essential for trophozoite survival, being the primary carbohydrate source for *B. coli* [25,26,27]. In fact, natural infections appear to be associated with carbohydrate-rich diets [9]. Since the medium composition proposed by Rees [34], rice starch has commonly been used in culture media. In our study, we used potato starch, which also proved effective, as trophozoites were observed feeding on it (Figure 2I, Appendix A). Due to the short culture period, we did not assess its long-term suitability.

An important factor in excystation is the accompanying microbiota, which are considered essential for the development and maintenance of cultures [21] and have been present in virtually all studies [22,25,26,27,35,43,44,45]. In early studies, bacterial and fungal control was achieved by inoculating small amounts of fecal material, thereby introducing limited microbial content, and by transferring small volumes during subculturing [22,35]. Later studies introduced antibiotics and antifungals, such as penicillin, streptomycin, gentamicin, and amphotericin B [24,25,26,27,43,44,45]. As our goal was not to maintain long-term cultures but to investigate the excystation process, we avoided the use of antibiotics to preserve the natural microbiota. However, this may have affected not only the survival of excysted trophozoites due to nutrient depletion and medium acidification but also the long-term viability of the trophozoites within the cysts and their ability to initiate the excystation process.

Early researchers suggested that bacteria are required as a food source for the ciliate and that either their absence or overgrowth negatively impacts culture maintenance [22,35], a pattern also observed in cultures of rumen ciliates [20]. Auerbach’s experiments [43] indicated that bacterial excretion/secretion products, rather than live bacteria themselves, may be sufficient to support growth. In our study, bacteria (or their products) also appeared necessary for excystation, as all activated cysts contained bacteria internally. However, based on current data, it is not possible to determine their role, whether as a mechanical stimulus or as biochemical or nutritional factors. The complex composition of fecal material in culture prevents firm conclusions, and specific studies are required.

Culture temperature is another important factor, though available data remain contradictory. Some authors have reported that trophozoites can survive for several days in moist feces at room temperature [34,46], while others suggest survival is limited to only a few hours post-defecation [47]. In culture, trophozoites have been maintained below 30 °C [26,32,34,35,44,48]. Although growth may be more robust at 37 °C [44], survival is generally better at lower temperatures [26,44]. Cysts are thought to survive in the environment for several weeks [40], but no specific studies have assessed this. In our protocol, enzymatic digestion was performed at 37 °C (optimal for the enzymes used), but cultures were incubated at 28 °C post-digestion. In prior assays at 37 °C, most cysts contained morphologically altered trophozoites that stained with vital dyes, indicating non-viability. The failure of development at the host’s body temperature is surprising and remains unexplained. It is possible that during excystation, higher temperatures increase metabolic demand, and if trophozoites cannot acquire sufficient nutrients (due to a lack of bacterial ingestion because of a failure of cyst wall rupture), they may die within a few hours. The first hours in culture establishment are decisive, as in those cases in which no activated cysts were detected in the first 24 hpi, no later activations were observed.

The excystation protocol used was reproducible but affected by the initial viability of the cysts. In this study, samples always originated from the same animals and were processed identically; thus, the cause of viability variability remains unclear. These findings align with previous reports suggesting that the low success rate (<10%) of cultures initiated solely from cyst-containing feces may be partially explained by fluctuations in cyst viability [24,25]. It can be hypothesized that the host also plays a role in cyst viability and infectivity, as it may influence the parasite’s ability to excyst or survive in a new host. It is known that rRNA sequence variation can affect gene expression and phenotype [49]. It can be hypothesized that the molecular machinery of the dormant trophozoite within the cyst may not be fully compatible with a new host or, in culture, with the conditions of the in vitro system. Little is known about the extent of rRNA variation in *B. coli*, as it has only been documented for the ITS1 and ITS2 regions [29]. Further studies on the in vitro excystation and maintenance of this ciliate, including its genetic characterization, would help to shed light on this issue.

Cyst viability was assessed using vital dye exclusion. Both dyes employed (methylene blue and trypan blue) produced equivalent results. Trypan blue was used in the only previous study assessing *B. coli* cyst viability [50], while methylene blue was used here for the first time with this parasite; previously, it was used to evaluate *Acanthamoeba* trophozoite viability [51]. This compound is potentially toxic [52], but this is likely only relevant for prolonged exposure, longer than in our experiments. Congo red was also considered but ultimately discarded, as it may alter the chitin microfibrils of the cyst wall [53]. Additionally, since the DMEM base contained phenol red as a pH indicator, blue dyes provided greater contrast and facilitated the identification of non-viable cysts. Methylene blue produced stronger staining than trypan blue, aiding better differentiation under our conditions. With both dyes, we observed unstained cysts, suggesting that the intact cyst wall prevents dye penetration. In such cases, a fully stained cyst indicates a fissured wall and a non-viable cell, while an unstained cyst implies wall integrity. This does not necessarily mean there is a viable trophozoite inside the cyst. Indeed, we observed unstained cysts containing morphologically altered trophozoites. This highlights the challenge of distinguishing viable (containing live, potentially infective cells) from infective (capable of establishing infection) cysts. Although a combination of morphology and dye exclusion may identify viable cysts, no current method can determine infectivity with certainty.

## 5. Conclusions

The aim of this study was not to develop an optimized protocol for the excystation of *B. coli* but rather to identify and describe the sequential steps involved in cyst activation and excystation. Nevertheless, the protocol appears to improve the success rate of culture establishment, although this requires validation through further experiments. Importantly, we identified several factors that may play a key role in this process and should be investigated further in targeted studies to improve excystation rates. A more efficient excystation protocol could provide a valuable foundation for future research on cyst viability and the evaluation of chemotherapeutic agents.

The in vitro excystation of *B. coli* from fecal samples requires an initial rupture of the cyst wall, which is followed by bacterial entry that appears to activate the enclosed trophozoite. Mechanical disruption of the weakened cyst wall subsequently allows the trophozoite to emerge. The initial rupture may result from a combination of factors, including enzymatic digestion and likely mechanical interactions with other intestinal components. Final emergence is probably facilitated by both these mechanical interactions and the internal deformations caused by trophozoite movement within the cyst. Future studies are needed to assess the relative importance of these mechanical and biochemical factors in the excystation process.

In addition to their possible role as mechanical agents in cyst rupture, starch granules are also important as a carbohydrate source for sustaining cultures. Rice starch has traditionally been used in media formulations, but our results indicate that potato starch is also suitable. It is likely that, as with the base culture medium, various starch sources can support the in vitro culture of *B. coli*.

## Figures and Tables

**Figure 1 pathogens-14-00725-f001:**
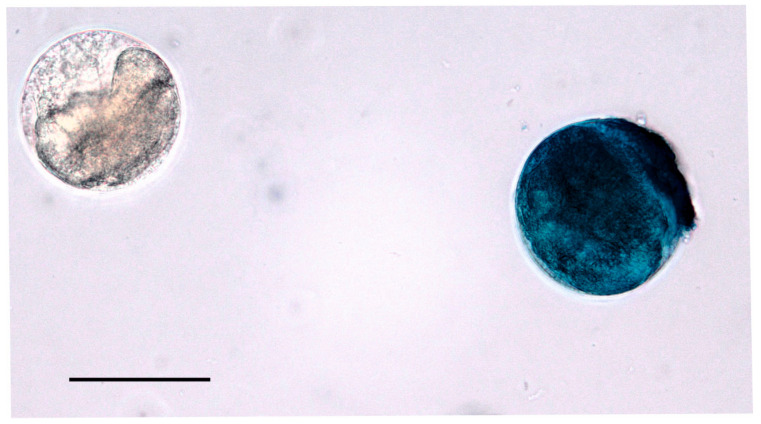
Determination of cyst viability using methylene blue vital staining. The cyst on the left appears colorless and is considered viable, whereas the cyst on the right is fully stained blue and considered non-viable. Scale bar: 50 µm.

**Figure 2 pathogens-14-00725-f002:**
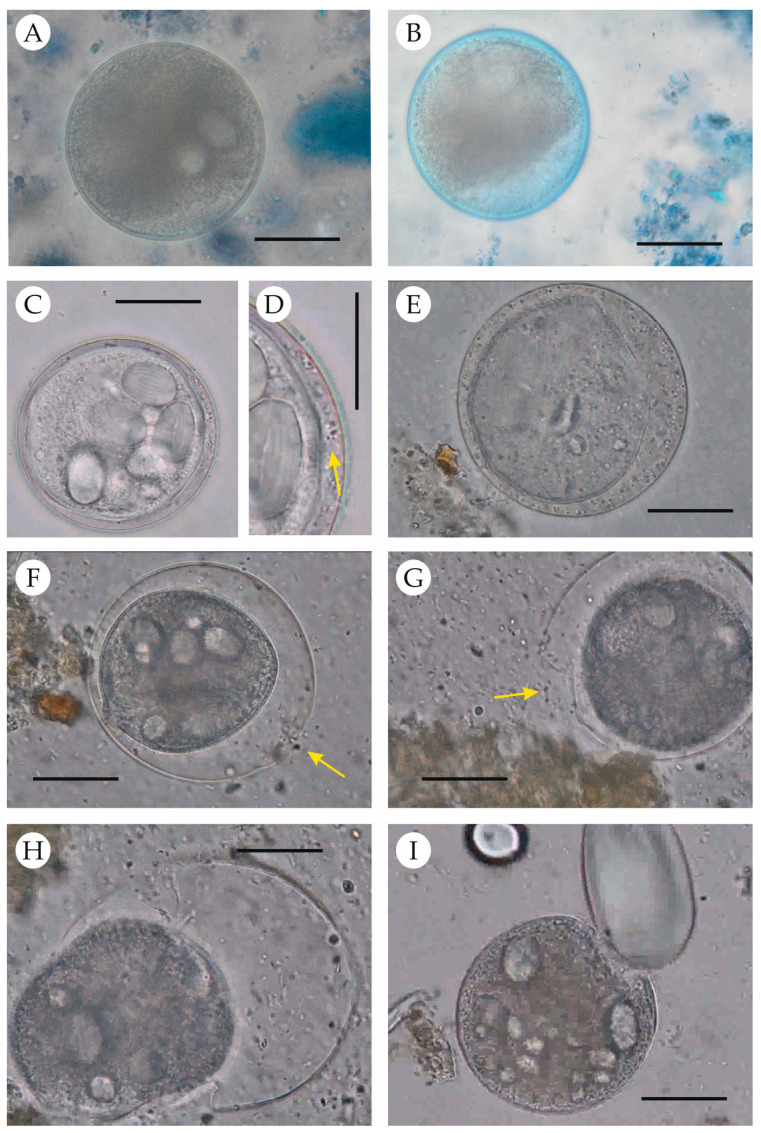
Progression of the excystation process in *Balantioides coli*. (**A**) Cyst following enzymatic digestion. (**B**) Cyst with a fissure (usually not observable), allowing vital stains to penetrate the intracystic space. (**C**) Cyst with bacteria present in the intracystic space. (**D**) Close-up showing bacteria (granular appearance; arrow) within the intracystic space. (**E**) Trophozoite is moving and causes distension of the cyst wall, resulting in an expanded intracystic space filled with bacteria. (**F**) Distension and deformation of the cyst wall; arrow indicates the rupture site. (**G**) Enlargement of the rupture in the cyst wall (arrow). (**H**) Moment of trophozoite emergence from the cyst. (**I**) Free trophozoite feeding on a starch granule. Scale bar (all images): 20 µm.

## Data Availability

The original contributions presented in the study are included in the article or accessible via the Appendix A. Further inquiries can be directed to the corresponding author.

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
