# Peer review of "Challenges and Achievements in the In Vitro Culture of Balantioides coli: Insights into the Excystation Process"

_pathogens, 2025, doi:10.3390/pathogens14080725_

Round 1

Reviewer 1 Report

Comments and Suggestions for Authors

Balantioides coli is a ciliated intestinal parasite in humans and animals. Excystation is its necessary initial stage of cyst infection in hosts. Conducting specific study on this process has significant theoretical importance for understanding the pathogenesis of B. coli. This study simulated the natural excystation in vivo to reveal some critical factors for both cyst infection and in vitro cultivation establishments. The results showed that wall rupture and bacterial entry were critical for activation in excystation of B. coli.

However, there are some problems in the manuscript as follows:

  1. The author believes that the addition of pepsin and trypsin during the process of excystation is an important influencing factor. However, the excystation situation without adding these two digestive enzymes was not observed in this manuscript. These results should be showed in the revised MS.
  2. Furthermore, the author believes that the entry of bacteria is another crucial factor in activating the trophozoite and ultimately causing its excystation. These findings seem less convincing. Will the situation of excystation decrease significantly after the addition of antibiotics? There is no data to support this statement.
  3. The commercialized DMEM cell culture medium usually contains L-glutamine. In the in vitro excystation test, why did the authors add L-glutamine as an extra component?
  4. Why did the author choose DMEM medium instead of other commonly used media such as Pavlova? If the excystating results of cysts of B. coli in various culture media are compared, the results will be more convincing and applicable.

Author Response

First of all, we would like to thank the reviewer for the time dedicated to reading and commenting on our manuscript. In response to the comments provided, we would like to clarify the following:

Comment 1:

  1. The author believes that the addition of pepsin and trypsin during the process of excystation is an important influencing factor. However, the excystation situation without adding these two digestive enzymes was not observed in this manuscript. These results should be showed in the revised MS.
  2. Furthermore, the author believes that the entry of bacteria is another crucial factor in activating the trophozoite and ultimately causing its excystation. These findings seem less convincing. Will the situation of excystation decrease significantly after the addition of antibiotics? There is no data to support this statement.

Response 1:

As stated in the abstract (lines 17–19), the aim of the present study is to describe the sequence of events involved in excystation and to identify factors potentially important for culture establishment. The objective is not to optimize the excystation procedure, but rather to describe the excystation process itself. It is true that, in the Introduction (lines 76–77 in the former manuscript), the objective may not be as clearly stated as in the abstract, and the phrase “to investigate the process of excystation” may be interpreted as an investigation into the relevance of the various factors involved in the process. We have modified the Introduction to clarify the objective of the study (lines 76–78 in the revised version of the manuscript):

“To date, no studies have been conducted on the development process of cysts or on the events occurring during the initial hours of cultivation. Accordingly, the aim of the present study is to describe the sequence of events involved in excystation and identify factors potentially important for culture establishment.”

Please note, this is an observational study that has allowed us to identify certain elements that may be important for the success of the excystation process and the establishment of cultures. The challenging aspects pointed out by the reviewer (enzyme treatment, antibiotics), as well as others (media composition, pH, physical cyst disruption, bacteria, temperature, initial cyst viability), are comented in the Discussion section. Each of these factors should be the subject of specific and detailed studies, and the improvements resulting from such investigations would be of great value not only for culture establishment but also for environmental and epidemiological studies (e.g., for evaluating cyst infectivity from a given water or food source).

Regarding the two specific points raised by the reviewer, and considering the aim of the study, we did not seek to determine whether both enzymatic digestion steps are necessary to activate the cysts, or whether they would be more effective at different durations, concentrations of pepsin or trypsin, or temperatures. Rather, since both digestions occur in the host’s intestine during the early phases of digestion (presumably when the excystation of the parasite also takes place) we subjected the cysts to these enzymatic environments in an attempt to reasonably simulate in vitro the conditions that occur in vivo.

Regarding the role of bacteria, we note in the Discussion section (lines 273–293) that their importance in excystation (and even in culture maintenance) remains unclear. Despite being considered essential for the development and maintenance of cultures, it is not possible to determine their role in the excystation process, whether as a mechanical stimulus or as biochemical or nutritional factors.

Comment 2:

  1. The commercialized DMEM cell culture medium usually contains L-glutamine. In the in vitro excystation test, why did the authors add L-glutamine as an extra component?

Response 2: Multiple DMEM formulations are available from different manufacturers, each offering variants that may differ in supplementary components such as glucose, sodium bicarbonate, sodium pyruvate, or L-glutamine. In our case, the formulation used did not include L-glutamine, which was therefore added manually.

Comment 3:

  1. Why did the author choose DMEM medium instead of other commonly used media such as Pavlova? If the excystating results of cysts of B. coli in various culture media are compared, the results will be more convincing and applicable.

Response 3: DMEM was selected because it is one of the culture media reported in the literature as suitable for B. coli. As previously stated, our aim was not to evaluate excystation success rates under varying protocol conditions, but to provide a descriptive account of the process.

Reviewer 2 Report

Comments and Suggestions for Authors

Dear Authors,

I would like to congratulate you on your work and emphasize the relevance of this study for the development of future research on Balantioides coli, a highly neglected protozoan. This study addresses the processes, challenges, and achievements related to the excystation of B. coli during the establishment of in vitro cultivation of this protozoan from orangutan samples. It has yielded innovative and highly relevant results for improving the inoculum and maintaining B. coli in vitro, particularly from cystic forms, which show a lower success rate compared to direct isolation from trophozoites.

Please find below my comments:

Methodology

  • Line 80: Please indicate the total number of collected samples.
  • Line 119: The portion of the sample used for molecular analysis is somewhat unclear. Could you rephrase this part? Was the material used for PCR to confirm the B. coli taxon the sediment obtained after centrifugation at 1500 rpm for 2 minutes?
  • Line 125: Was the DNA extracted from only one sample, or from all six that had at least 75% viable cysts and were later analyzed? Please specify.
  • Line 138: This is a rather rich medium, possibly more expensive and complex to prepare compared to others previously used for B. coli cultivation. Was there a specific reason for choosing this medium instead of a simpler liquid one, such as Pavlova or TYSM-9?
  • Line 150: Is the suspension used at this stage the one stored in 50 mL tubes? Since your study includes several steps and sample divisions for various protocols, clarifying this point in the text would be helpful.
  • Line 160: In this study, antibiotics were not added in order to preserve the sample's accompanying microbiota. However, other studies use antibiotic solutions to control bacterial populations and prevent faster acidification of the medium, which could lead to parasite death. Do you think this choice may have influenced the duration of parasitic viability? It would be worthwhile to include this consideration in the discussion. It would also have been very interesting if some taxa of the accompanying microbiota, which was preserved, could have been identified through molecular biology.
  • Line 166: Please indicate the magnification at which the video was recorded.

Results and Discussion

  • Figure 1: Could you provide a TIFF file with higher resolution? The letters indicating parts of the image appeared low in quality in the PDF version of the article, and the image was not included among the files uploaded to the journal platform—only the individual images and the video are available. If you wish to retain the panel with the composite images, please provide the exact file.
  • Line 169: Please include the total number of samples analyzed.
  • Line 171: Did you determine the B. coli variant? If so, please specify. A recently published study observed clear polymorphism among trophozoites from Old World primates (variant A) and isolates from Neotropical primates (variant B). Differences were also found in the success and duration of in vitro cultivation between these isolates. I believe that greater adaptation of the parasite to the gut environment of Old World primates might influence the viability of parasitic forms. What is your opinion? Highlighting the potential influence of host-related factors—such as diet, anatomy, and behavior—on parasite viability is a valuable hypothesis and could enrich the discussion.
  • Line 198: Balantioides coli should be italicized.
  • Line 286: It is interesting how the optimal temperatures for B. coli cultivation remain controversial. I believe that temperatures between 36°C and 37°C are most suitable for trophozoite forms, as they mimic the body temperature of pigs—considered reservoirs—and non-human primates. However, cysts, being environmentally resistant forms, are capable of remaining viable at lower temperatures. What are your thoughts on this? These points could be further addressed in the discussion.

Conclusion

  • Line 327: Although developing a protocol to stimulate B. coli excystation was not the primary aim of your study, your methodology was successful, and I believe this should be more strongly emphasized in the text. Of course, further analysis is needed to optimize the process, but this is already a significant step forward.

Author Response

REVIEWER 2

First of all, we would like to thank the reviewer for the time dedicated to reading and commenting on our manuscript. The comments and reflections provided are very insightful and will be taken into consideration for future studies on the culture of B. coli.

With respect to the observations on the manuscript, we would like to state the following:

Comment 1: Methodology

Line 80: Please indicate the total number of collected samples.

Response 1: We have added the number as requested (line 81): “Ten fecal samples were collected from orangutans …”

Comment 2: Line 119: The portion of the sample used for molecular analysis is somewhat unclear. Could you rephrase this part? Was the material used for PCR to confirm the B. coli taxon the sediment obtained after centrifugation at 1500 rpm for 2 minutes?

The analysis was made using the sediment obtained after the concentration technique. We have modified the text to make it clearer (lines 122-123): “Twenty microliters of the sediment obtained from that sample by the acetate-ether concentration method were diluted …”

Comment 3: Line 125: Was the DNA extracted from only one sample, or from all six that had at least 75% viable cysts and were later analyzed? Please specify.

Response 3: No other compatible ciliates have been reported in orangutans in the bibliography (Buxtonella sp. has been cited only in cercopithecid primates), and genetic analyses carried out in previous years on fecal samples from the orangutans at the ZooAquarium (as part of undergraduate and postgraduate student projects) have consistently yielded the same result: B. coli. In any case, to ensure that the parasite used in the present study was indeed B. coli, we repeated the genetic analysis on one of the samples. Given the existing background (both from the literature and from our own analysis history), we considered it unnecessary to test all samples individually.

We have slightly modified the text (lines 121–122) as follows: “Genetic analysis was performed using one of the selected samples employed in the assays."

Comment 4: Line 138: This is a rather rich medium, possibly more expensive and complex to prepare compared to others previously used for B. coli cultivation. Was there a specific reason for choosing this medium instead of a simpler liquid one, such as Pavlova or TYSM-9?

Response 4: DMEM is one of the media considered suitable for culture, and given its complex composition, we assumed it would not be a limiting factor in the excystation process. Future studies are needed to determine, among many other parameters, which medium may be most suitable for excystation, acknowledging that it may not necessarily be the same as that used for trophozoite culture.

Comment 5: Line 150: Is the suspension used at this stage the one stored in 50 mL tubes? Since your study includes several steps and sample divisions for various protocols, clarifying this point in the text would be helpful.

Response 5: We obtained a concentrated sediment after acetate-ether concentration to check for the presence of cysts (Section 2.1, lines 96–98) and to perform the genetic analysis (Section 2.3, lines 120–121); and a sample suspension (original sample in distilled water), which was used for cyst quantification and viability assessment (Section 2.2), as well as for the excystation procedure (Section 2.5). To clarify this, we have modified the text as follows (lines 101–103):
“This suspension was kept at room temperature and used as the starting material for subsequent analyses (Sections 2.2 and 2.5) conducted in this study.”

Comment 6: Line 160: In this study, antibiotics were not added in order to preserve the sample's accompanying microbiota. However, other studies use antibiotic solutions to control bacterial populations and prevent faster acidification of the medium, which could lead to parasite death. Do you think this choice may have influenced the duration of parasitic viability? It would be worthwhile to include this consideration in the discussion. It would also have been very interesting if some taxa of the accompanying microbiota, which was preserved, could have been identified through molecular biology.

Response 6: This is a very insightful comment and may indeed be highly relevant to the excystation process. The objective of this study was not to determine the optimal conditions for excystation, but rather to describe the process and to highlight potential key factors, and this may be one of them. We have added the following sentence to the manuscript (lines 281–283):
“However, this may have affected not only the survival of excysted trophozoites due to nutrient depletion and medium acidification, but also the long-term viability of the trophozoite within the cyst and its ability to initiate the excystation process.”

Comment 7: Line 166: Please indicate the magnification at which the video was recorded.

Response 7: We have added it (line 170): “with 100 × objective lens”.

Comment 8: Results and Discussion. Figure 1: Could you provide a TIFF file with higher resolution? The letters indicating parts of the image appeared low in quality in the PDF version of the article, and the image was not included among the files uploaded to the journal platform—only the individual images and the video are available. If you wish to retain the panel with the composite images, please provide the exact file.

Response 8: Please note that images A and B in Figure 2 (the panel with the composite images) are original photographs, whereas the other images are still frames extracted from the video. All images provided (A, B, and the composite) have a resolution of 600 dpi. If necessary, and should the article be accepted, we will be happy to provide higher-resolution images or alternative file formats upon the editor’s request.

Comment 9: Line 169: Please include the total number of samples analyzed.

The total number of samples analyzed (10) is given in line 81. “All fecal samples” refers to all of them (the 10 analyzed). Anyway, we have modified the text: “All ten fecal samples analyzed were …”

Comment 10: Line 171: Did you determine the B. coli variant? If so, please specify. A recently published study observed clear polymorphism among trophozoites from Old World primates (variant A) and isolates from Neotropical primates (variant B). Differences were also found in the success and duration of in vitro cultivation between these isolates. I believe that greater adaptation of the parasite to the gut environment of Old World primates might influence the viability of parasitic forms. What is your opinion? Highlighting the potential influence of host-related factors—such as diet, anatomy, and behavior—on parasite viability is a valuable hypothesis and could enrich the discussion.

Response 10: We did not explicitly state the B. coli variant, but it can be identified in Supplementary File 1 as A0 (characterized by three ‘A’ nucleotides at positions 42–45 and three ‘T’ nucleotides at positions 49–51). We have now added this information to the manuscript (lines 174–175):
“Genetic analyses confirmed the species as B. coli, sequence variant A0, showing …”

We have searched scientific databases but were unable to locate the study mentioned by the reviewer regarding differential ribosomal gene expression in B. coli depending on the host. We would be very grateful if the reviewer could kindly share the reference with us.

Regarding the importance of the host in parasite viability, this is a complex topic that warrants dedicated investigation. We distinguish between viability and infectivity; the former refers to the parasite being alive, while the latter implies that it is both alive and capable of develop in a new host. We agree with the reviewer that the host may influence the gene expression of B. coli; however, rather than affecting its viability, we believe it is more likely to impact its infectivity. For some species, it is known that they can express different copies of rRNA to produce slightly distinct ribosomes adapted to specific environmental conditions (see, e.g., https://royalsocietypublishing.org/doi/10.1098/rstb.2023.0379). This phenomenon has not yet been studied in B. coli, but it can be hypothesized that the parasite may modulate its gene expression depending on the host species (i.e., in response to the biochemical environment of the host's gut, which is influenced by diet, microbiota composition, enzymatic profile, and other factors). In B. coli, a cyst is essentially a trophozoite in a dormant state, but it retains the molecular components (i.e., rRNA) that were expressed during its active phase in the host intestine. It is possible that, even if the cyst or trophozoite is viable (i.e., alive), its “active” molecular machinery may not be well suited to the new host, thereby limiting its ability to establish infection or persist. It can be also hypothesized that a similar situation could occur in in vitro cultures, where certain allelic variants may be better adapted to the artificial conditions than others.

Following the reviewer’s suggestion, we have included a comment on the possible influence of the original host on the culture success (lines 314-322): “It can be hypothesized that the host also plays a role in cyst viability and infectivity, as it may influence the parasite’s ability to excyst or survive in a new host. It is known that rRNA sequence variation can affect gene expression and phenotype [51]. It can be hypothesized that the molecular machinery of the dormant trophozoite within the cyst may not be fully compatible with a new host or, in culture, with the conditions of the in vitro system. Little is known about the extent of rRNA variation in B. coli, as it has only been documented for the ITS1 and ITS2 regions [29]. Further studies on the in vitro excystation and maintenance of this ciliate, including its genetic characterization, would help to shed light on this issue.”

Comment 11: Line 198: Balantioides coli should be italicized.

Response 11: Corrected, thanks for pointing it out this error.

Comment 12: Line 286: It is interesting how the optimal temperatures for B. coli cultivation remain controversial. I believe that temperatures between 36°C and 37°C are most suitable for trophozoite forms, as they mimic the body temperature of pigs—considered reservoirs—and non-human primates. However, cysts, being environmentally resistant forms, are capable of remaining viable at lower temperatures. What are your thoughts on this? These points could be further addressed in the discussion.

Response 12: We believe this point is addressed in the Discussion (lines 293–308). For the purposes of the present study, the main question is not the temperature at which cysts remain viable for longer periods, but rather the temperature at which excystation and culture would take place (presumably corresponding to that of the host). However, as stated in that paragraph, findings from different studies are contradictory, and there is no clear or definitive explanation for the parasite’s growth at different temperatures, with some reports indicating an inability to develop at 37 °C.

Comment 13: Conclusion. Line 327: Although developing a protocol to stimulate B. coli excystation was not the primary aim of your study, your methodology was successful, and I believe this should be more strongly emphasized in the text. Of course, further analysis is needed to optimize the process, but this is already a significant step forward.

Response 13: We have modified the text (lines 344-346) as follows: “Nevertheless, the protocol appears to improve the success rate of culture establishment, although this requires validation through further experiments. Importantly, we have identified several factors …”

Reviewer 3 Report

Comments and Suggestions for Authors

Interesting article focusing mainly on the details of in vitro culture  and excystation process of Balantioides coli.

The specific comments are listed below:

Abstract
Line 24: “This appeared to stimulate the trophozoites. Increased motility led to progressive wall rupture until emergence occurred.” — This sentence needs clarification.
It should be revised to specify that the trophozoites are being stimulated to increase their motility:

This appeared to stimulate the trophozoites, whose increased motility progressively weakened and ruptured the wall, allowing their emergence.

Introduction
Line 47: “… reservoir of the parasite .” — Should be corrected to: “… reservoir of this parasite .”

Results
Line 169: “All fecal samples analyzed were macroscopically normal.” — This sentence needs clarification. It would be helpful to specify what “normal” refers to (e.g., consistency, color, presence of visible parasites, etc.).

Comments on the Quality of English Language

Introduction
Line 51: “… up to 100 μm 50 in length and cysts may exceed …” — A comma should be inserted:

“… up to 100 μm in length, and cysts may exceed …”

Line 67: “The methods developed over the past decade are modifications of earlier procedures, employing different saline solutions as the base medium or varying sera as sources of proteins and amino acids.” — This sentence should be rephrased for clarity and readability:

“The methods developed over the past decade represent modifications of earlier procedures, utilizing different saline solutions as base media and employing various sera as sources of proteins and amino acids.”

Author Response

REVIEW 3

First of all, we would like to thank the reviewer for the time dedicated to reading and commenting on our manuscript. In response to the comments provided, we would like to response the following:

Comment 1: Abstract. Line 24: “This appeared to stimulate the trophozoites. Increased motility led to progressive wall rupture until emergence occurred.” — This sentence needs clarification.
It should be revised to specify that the trophozoites are being stimulated to increase their motility:

This appeared to stimulate the trophozoites, whose increased motility progressively weakened and ruptured the wall, allowing their emergence.

Response 1: Done as requested. Thanks very much for suggesting the alternative text.

Comment 2: Introduction. Line 47: “… reservoir of the parasite .” — Should be corrected to: “… reservoir of this parasite .”

Response 2: Done as requested.

Comment 3: Results. Line 169: “All fecal samples analyzed were macroscopically normal.” — This sentence needs clarification. It would be helpful to specify what “normal” refers to (e.g., consistency, color, presence of visible parasites, etc.).

Response 3: We have changed the text as follows (lines 172-174): “…were macroscopically normal, with formed consistency, normal coloration, and no visible mucus, excess liquid, or apparent parasitic forms”.

Comment 4: Comments on the Quality of English Language. Introduction. Line 51: “… up to 100 μm 50 in length and cysts may exceed …” — A comma should be inserted: “… up to 100 μm in length, and cysts may exceed …”

Response 4: Done as suggested.

Comment 5: Line 67: “The methods developed over the past decade are modifications of earlier procedures, employing different saline solutions as the base medium or varying sera as sources of proteins and amino acids.” — This sentence should be rephrased for clarity and readability:

“The methods developed over the past decade represent modifications of earlier procedures, utilizing different saline solutions as base media and employing various sera as sources of proteins and amino acids.”

Response 5: Done as requested. Again, thanks for suggesting the alternative text.